# Loss of Nitric Oxide Induces Fibrogenic Response in Organotypic 3D Co-Culture of Mammary Epithelia and Fibroblasts—An Indicator for Breast Carcinogenesis

**DOI:** 10.3390/cancers13112815

**Published:** 2021-06-05

**Authors:** Gang Ren, Xunzhen Zheng, Vandana Sharma, Joshua Letson, Andrea L. Nestor-Kalinoski, Saori Furuta

**Affiliations:** 1Department of Cancer Biology, College of Medicine and Life Sciences, University of Toledo Health Science Campus, 3000 Arlington Ave., Toledo, OH 43614, USA; gang.ren@utoledo.edu (G.R.); Xunzhen.Zheng@utoledo.edu (X.Z.); vandana.sharma@utoledo.edu (V.S.); Joshua.Letson@rockets.utoledo.edu (J.L.); 2Department of Surgery, College of Medicine and Life Sciences, University of Toledo Health Science Campus, 3000 Arlington Ave., Toledo, OH 43614, USA; Andrea.Kalinoski@utoledo.edu

**Keywords:** organotypic 3D co-culture, extracellular matrix, mammary gland, fibroblasts, myofibroblasts, desmoplasia, mammary epithelial cells, arginine, nitric oxide, mammary morphogenesis

## Abstract

**Simple Summary:**

Fibrosis, which is often caused by chronic diseases and environmental substances, is closely associated with cancer. Thus, the development of a robust method allowing for deep studies of the linkage between fibrosis and cancer is essential. Here, we tested whether our novel three-dimensional (3D) co-culture of breast epithelia and fibroblasts would be a suitable model for that purpose. We compared the phenotypic effects of L-NAME, an inhibitor of nitric oxide (NO) production, on 3D mono- and co-cultures. We previously reported that prolonged NO depletion with L-NAME caused fibrosis and tumorigenesis in mouse mammary glands. Such in vivo effects of L-NAME were well recapitulated in 3D co-cultures, but not in 3D mono-cultures of epithelia and fibroblasts. These results support not only the essential roles of the presence of the stroma in cancer development, but also the utility of this co-culture in studying the causal relationship between fibrosis and cancer.

**Abstract:**

Excessive myofibroblast activation, which leads to dysregulated collagen deposition and the stiffening of the extracellular matrix (ECM), plays pivotal roles in cancer initiation and progression. Cumulative evidence attests to the cancer-causing effects of a number of fibrogenic factors found in the environment, diseases and drugs. While identifying such factors largely depends on epidemiological studies, it would be of great importance to develop a robust in vitro method to demonstrate the causal relationship between fibrosis and cancer. Here, we tested whether our recently developed organotypic three-dimensional (3D) co-culture would be suitable for that purpose. This co-culture system utilizes the discontinuous ECM to separately culture mammary epithelia and fibroblasts in the discrete matrices to model the complexity of the mammary gland. We observed that pharmaceutical deprivation of nitric oxide (NO) in 3D co-cultures induced myofibroblast differentiation of the stroma as well as the occurrence of epithelial–mesenchymal transition (EMT) of the parenchyma. Such in vitro response to NO deprivation was unique to co-cultures and closely mimicked the phenotype of NO-depleted mammary glands exhibiting stromal desmoplasia and precancerous lesions undergoing EMT. These results suggest that this novel 3D co-culture system could be utilized in the deep mechanistic studies of the linkage between fibrosis and cancer.

## 1. Introduction

The formation of the dense collagenous stroma, namely, desmoplasia, is a hallmark of many types of carcinomas, especially breast cancer, and is responsible for the hard “lump” appearance of tumors by palpation [1]. Desmoplasia was initially thought to be the result of the condensation of pre-existing collagen fibers [2]. However, it was later found that this phenomenon is primarily due to the enhanced collagen biosynthesis by myofibroblasts, a subpopulation of cancer-associated fibroblasts (CAFs), coming to populate the tumor interstitium to augment tumor growth [3,4,5]. Myofibroblasts are formed when locally residing fibroblasts, epithelia, endothelia and other types of cells differentiate into highly contractile (due to high levels of α-smooth muscle actin (α-SMA)) and secretory cells that primarily engage in tissue repair upon wounding [6,7]. Excessive myofibroblast activation, however, could lead to their uncontrolled contraction and exorbitant secretion of the extracellular matrix (ECM) and mitogenic factors (such as hepatocyte growth factor (HGF)) [4,8]. This could ultimately result in tissue fibrosis and hypertrophic scars that destroy organ functions [6]. The abundant myofibroblasts in the tumor stroma thus greatly contribute to the pathogenesis of tumors [9] and their prevalent phenotype referenced as “wounds that never heal” [10].

Myofibroblast differentiation in the tumor stroma is primarily induced by paracrine signaling of growth factors secreted by tumor cells and immune cells as well as by injured microvasculature. These factors include transforming growth factor β (TGF-β), platelet-derived growth factor (PDGF) and vascular endothelial growth factor (VEGF) [11,12,13,14,15,16]. These myofibroblasts in the tumor stroma secrete pro-tumor factors (such as TGF-β) as well as fibrous collagen and collagen cross-linking enzymes to make the ECM stiff. All these products of myofibroblasts work together to promote tumor progression and invasion [17,18,19]. Importantly, myofibroblasts could already be detected in precancerous lesions or tumor-adjacent normal tissues, leading to a notion that they also play roles in tumor initiation [5,20,21,22]. Accordingly, the emergence of myofibroblasts in non-cancerous tissues could indicate their cancer predisposition, which would be utilized in screening for potential carcinogenic agents.

Previously, we reported that pharmaceutical deprivation of the endogenous nitric oxide (NO) in wild-type female mice at prepubertal to pubertal stages induced precancerous mammary lesions. These lesions were characterized by desmoplastic stroma and intraductal papillomas that highly expressed TGF-β and HER2 oncogene [23]. In the present report, we tested whether such fibrogenic and carcinogenic effects of NO deprivation could be recapitulated in our novel organotypic 3D co-cultures [24]. Through this study, we examined whether this in vitro co-culture system could exhibit desmoplastic responses toward a known fibrogenic and carcinogenic factor to test its utility in the deep studies of the causal linkage between fibrosis and cancer.

This new organotypic 3D co-culture system utilizes the discontinuous ECM to model the complexity of glandular tissues, such as mammary glands [24]. The mammary gland is the arboreal structure extended from the nipple towards the terminal lobes composed of mammary acini—the milk producing units. Each acinus is composed of the apical layer of the milk-producing luminal epithelial cells (positive for cytokeratin 8/18 (CK8/18) and a Golgi apparatus marker GM130), and the basal layer of contractile myoepithelial cells (positive for cytokeratin 14 (CK14) and α6 integrin). Each acinus is surrounded by the laminin-rich basement membrane (BM) and then embedded in the collagen I-rich connective tissue harboring stromal cells (interstitium). Here, the BM serves as the barrier to separate the epithelium from the stromal connective tissue by forming stable epithelial-to matrix adhesions (Figure 1A,B) [25,26]. In our 3D co-culture, mammary epithelial cells and fibroblasts are separately seeded in physiologically relevant matrices: laminin-rich BM for epithelia vs. collagen I-rich membrane for fibroblasts. Both cell types interact through paracrine signaling, but without direct physical interactions [24]. This system is different from the original 3D organotypic co-culture system developed for surface epithelia (such as skin and respiratory tracts) which utilizes the continuous ECM that resembles the collagen-rich interstitial matrix (Figure 2A,B) [27,28,29]. Our co-culture system is also different from another type of 3D organotypic co-culture system, where the mixture of epithelia and stromal cells is cultured in the continuous matrix that resembles the laminin-rich BM [30,31].

We demonstrate here that the deprivation of the endogenous NO in 3D co-cultured mammary epithelia and fibroblasts induced myofibroblast differentiation owing to the upregulation of TGF-β signaling. NO deprivation in 3D co-cultures also induced epithelial–mesenchymal transition (EMT)—the emergence of the invasive phenotype [32]. Such in vitro phenotype closely mimicked the phenotype of mammary glands in vivo after NO deprivation [23] and was unique to 3D co-cultures. Our result suggests that this new co-culture system could recapitulate the physiological responses of glandular tissues toward fibrogenic and carcinogenic agents and could possibly be utilized in studying the causal linkage between fibrosis and cancer.

## 2. Materials and Methods

### 2.1. Antibodies

The following antibodies were used: anti-integrin α6 (BD Biosciences, Franklin Lakes, NJ, USA, Cat. # 555734); anti-GM130 (IF: Cell Signaling Technology, Danvers, MA, USA, Cat. # 12480S; IHC: Novus Biologicals, Littleton, CO, USA, Cat. # NBP1-89756); anti-human CK 14 (Thermo Fisher, Waltham, MA, USA, Cat. # MA5-11599); anti-human CK 18 (Thermo Fisher, Waltham, MA, USA, Cat. # PA514263); anti-mouse CK 14 (BioLegend, San Diego, CA, USA, Cat. # 905301); anti-mouse CK 8/18 (DSHB, Iowa City, IA, USA, Cat. # Troma-I); anti-alpha smooth muscle actin (Abcam, Cambridge, UK, Cat. # Ab7817); anti-E-cadherin (BD Biosciences, Franklin Lakes, NJ, USA, Cat. # 610182); anti-vimentin (Life Technologies, Carlsbad, CA, USA, Cat. # PA5-86264); anti-PAR3 (Millipore, Burlington, MA, USA, Cat. # 07-330); anti-laminin alpha3 (Novus Biologicals, Littleton, CO, USA, Cat. # CL3112); and phospho-SMAD3 antibody (pS423/pS425, Novus Biologicals, Littleton, CO, USA, Cat. # NBP1-77836SS).

### 2.2. Cell Lines

MCF10A human MECs were obtained from Barbara Ann Karmanos Cancer Institute (Detroit, MI, USA), and primary human mammary fibroblasts (MFs) from ScienCell (ScienCell Research Laboratories, Carlsbad, CA, USA, Cat. # 7630). All cell lines were authenticated by the suppliers and obtained with the appropriate Material Transfer Agreement. Mycoplasma testing gave negative results.

### 2.3. Cell Culture

MCF10A human MECs were cultured in DMEM/F12 (Thermo Fisher, Waltham, MA, USA, Cat. # 11320033) supplemented with 5% horse serum (Thermo Fisher, Waltham, MA, USA, Cat. # 16050122); 1% penicillin/streptomycin; 10 μg/mL insulin (Sigma, St. Louis, MO, USA, Cat. # I-1882); 20 ng/mL EGF (Sigma, St. Louis, MO, USA, Cat. # E-9644); 0.5 μg/mL hydrocortisone (Sigma, St. Louis, MO, USA, Cat. # H-0888); and 100 ng/mL cholera toxin (Sigma, St. Louis, MO, USA, Cat. # C-8052) [33]. Primary human MFs were cultured in Fibroblast Medium (FM, ScienCell Research Laboratories, Carlsbad, CA, USA, Cat. #2301) supplemented with 1% Fibroblast Growth Supplement (FGS, ScienCell Research Laboratories, Carlsbad, CA, USA, Cat. #2352) and 1% penicillin/streptomycin as described [24].

### 2.4. D Mono-Culture of Mammary Epithelial Cells and Mammary Fibroblasts

For 3D mono-culture of normal mammary epithelial cells, MCF10A cells at the density of 2.5 × 10^4^ were seeded on top of 200 μL polymerized Matrigel (BD Biosciences, Franklin Lakes, NJ, USA, growth factor reduced) per well of a 4-well plate (1.9 cm^2^/well) and covered with the growth medium containing 4% Matrigel as described [33]. Cultures were maintained for 10–21 days with addition of fresh medium on alternate days [33]. For 3D mono-culture of mammary fibroblasts (MFs), cells were grown in Matrigel: collagen I (1:1) mixture in the same manner as 3D co-culture (see below) [24]. To determine the effects of the inhibition of NO production, 3D cultures were kept under the treatment of 2.5 mM L-NAME (NOS inhibitor, N_ω_-Nitro-L-arginine methyl ester hydrochloride, Sigma-Aldrich, St. Louis, MO, USA), vehicle control (PBS) or NO synthase agonist L-arginine (2.5 mM) for 2–3 weeks with medium change every other day [34].

### 2.5. D Organotypic Co-Culture

A custom stamp for micropatterning the ECM in a 12-well plate insert was constructed, and 3D organotypic co-culture of mammary epithelial cells and mammary fibroblasts was performed as described (please see the detailed protocols in Ren et al. [24]). Co-culture was treated with 2.5 mM L-NAME in comparison to L-arginine (2.5 mM) and vehicle control. Co-culture was processed for paraffin-embedding/sectioning and analyzed by immunohistochemistry [23].

### 2.6. Animal Studies

All animal experiments were performed in accordance with The Guide for the Care and Use of Laboratory Animals: Eighth Edition (National Research Council, National Academy Press, Washington, DC, USA, 2011) and under the protocol (NO. 108658) approved by the Institutional Animal Care and Use Committee (IACUC) of the University of Toledo, Toledo, OH. Three-week-old female BALB/c mice (n = 18) were purchased from the Jackson Laboratory (Bar Harbor, MN) and housed under a 12-h light-dark cycle with ad libitum access to regular chow and water. Starting at the age of 4 weeks old, mice received intraperitoneal injection of 100 µL of a drug (vehicle: PBS, L-arginine (20 mg/kg) or L-NAME (20 mg/kg)) every other day for 6 weeks. At the end of the treatment period, mice were euthanized, and inguinal (number 5) mammary glands were harvested and processed for paraffin-embedding and sectioning. Most paraffin sections were deparaffinized, hydrated and analyzed by immunohistochemistry. Other sections were analyzed by SHG imaging to visualize stromal collagen density (see below).

### 2.7. Immunohistochemistry

To determine the expression of specific markers, paraffin-embedded sections of mouse mammary glands or co-cultures were analyzed by immunohistochemistry. Briefly, sections were deparaffinized in xylene, hydrated in a series of graded alcohols (100%, 95%, 70% and 50%), and treated with antigen unmasking solutions (Vector Laboratories, Inc. Burlingame, CA, USA) or with Tris-EDTA Buffer (10 mM Tris Base, 1 mM EDTA Solution, 0.05% Tween 20, pH 9.0) heated to 95–100 °C in a pressure cooker. Sections were blocked with nonimmune goat or horse serum (depending on the host species (goat or horse, respectively) of secondary antibodies used for immunofluorescence staining), and processed for immunofluorescence staining as described below.

### 2.8. Immunofluorescence Staining and Imaging

Immunofluorescence staining/imaging was performed as described [34]. Briefly, sample slides were incubated with primary antibody diluted in IF buffer (0.1% BSA, 0.2% Triton-X 100, 0.05% Tween 20, 0.05% NaN3 in PBS) overnight at 4 degrees in a humidified chamber. After intensive washing (three times, 15 min each) in IF buffer, slides were incubated with fluorescence-conjugated secondary antibodies (Molecular Probes) diluted in IF buffer for 1 h at room temperature. After extensive washing in IF buffer, nuclei were counterstained with 0.5 ng/mL DAPI. After mounting with anti-fade solution, slides were visualized on Olympus IX70 microscope using CellSens software (Olympus Corp., Shinjuku City, Tokyo, Japan). Confocal fluorescence imaging for Second Harmonics Generation (SHG) technique [35] was performed on Leica Microsystems TCS SP5 multi-photon laser scanning confocal microscope using Suite Advanced Fluorescence *(*LAS AF*)* software (Leica Microsystems, Wetzlar, Germany)

### 2.9. Image Analysis

Quantification of fluorescence signal in micrographs was performed with ImageJ version 1.8 software (NIH) referring to the owner’s manual (http://imagej.net/docs/guide/146.html, accessed date 11 February 2021). Briefly, a region of interest (ROI) was determined in reference to an image of DAPI-stained nuclei or mammary acini. For each ROI, the average intensity per pixel (background intensity was subtracted) and shape descriptors (circularity, roundness and aspect ratio) were measured (refer to the ImageJ user manual: https://imagej.nih.gov/ij/docs/menus/analyze.html, accessed date 11 February 2021). For each sample group, at least 50 measurements were performed. The statistical significance of the data was further evaluated using Graphpad Prism Version 5 software (Graphpad Software Inc., San Diego, CA, USA; see statistics section).

### 2.10. Statistics

All samples were prepared in replicates (*n* ≥ 3 for in vitro experiments; *n* > 6 for in vivo experiments). The sample numbers were determined by power analysis with G*Power software (developed by Dr. Axel Buchner, University of Düsseldorf, Düsseldorf, Germany) ensuring the adequate statistical power (≥ 0.8), the error probability (≤ 0.05) and the estimated effect size of 0.8 based on the previous study [36]. Statistical significance of the difference of means was tested by two-tailed t-tests (parametric) using Graphpad Prism Version 5 software. *p*-values of 0.05 or less were considered significant. Average results of multiple experiments are presented as the arithmetic mean ± SEM.

## 3. Results

### 3.1. Novel Organotypic 3D Co-Culture of the Mammary Gland

We recently reported our novel organotypic 3D co-culture system for modeling the mammary gland (Figure 2B(b),C)[24]. This system is different from the 3D co-culture system originally developed for surface epithelia (Figure 2B(a))[29] or another co-culture system where the mixture of epithelia and stromal cells is cultured in the continuous matrix [30,31]. In this new method, mammary epithelial cells and mammary fibroblasts are co-cultured in distinct locales of the discontinuous ECM (laminin-rich BM for mammary epithelial cells vs. collagen I-rich interstitial membrane for mammary fibroblasts) generated with the help of a custom-made stamp (Figure 2C) [24]. This method allows for the growth of each cell type in the distinct, physiologically relevant matrix, while also allowing for the paracrine interactions, without direct interactions, between them.

### 3.2. Deprivation of NO Influences the Gross Phenotype of 3D Co-Cultures of Mammary Epithelial Cells and Fibroblasts in a Manner that Resembles the In Vivo Response

We previously demonstrated that the deprivation of NO in wild-type female mice with L-NAME, a competitive inhibitor of NO synthase (NOS) [36], induces precancerous mammary lesions surrounded by the desmoplastic ECM [23]. To test whether such fibrogenic effects of NO deprivation could be recapitulated in our new 3D co-culture system, we treated co-cultures with L-NAME in comparison to NOS agonist, L-arginine, and vehicle control (PBS). We also gave the same treatments to 3D mono-cultured mammary epithelial cells and 3D mono-cultured fibroblasts as the negative controls. We then compared the phenotypes of drug-treated 3D co-cultures and mono-cultures with that of mammary glands in vivo.

Histological examinations of drug-treated mammary tissues revealed that L-NAME treatment induced precancerous lesions characterized by the outgrowth of mammary epithelia in the ducts (intraductal papillomas) [23]. Such lesions had reduced circularity and were accompanied by the dense, fibroblast-enriched connective tissue around the glands (Figure 3A(a–c),B) [23]. In 3D mono-cultures of epithelia, L-NAME treatment also induced the formation of hypertrophic, elongated colonies (Figure 3A(d–f),B), consistent with our previous reports [23,34]. On the other hand, 3D mono-cultured fibroblasts did not show any morphological change after L-NAME treatment (Figure 3A(g–i)). This is likely due to the very low or no expression of NOS 1–3 in fibroblasts of normal mammary tissues, where NOS 1–3 expression is primarily focused on the epithelia [23]. On the other hand, in 3D co-cultures, L-NAME treatment induced hypertrophic, elongated morphologies of mammary epithelia, as well as dense, irregular arrangements of fibroblasts, in a manner that resembled the in vivo phenotype (Figure 3A(j–o),B). These results suggest that the phenotypic effects of L-NAME on fibroblasts require neighboring epithelia.

### 3.3. NO Deprivation Induces Fibrogenic Signals in 3D Co-Cultures of Mammary Epithelial Cells and Fibroblasts in a Manner that Resembles the In Vivo Response

To further confirm the fibrogenic effects of NO-deprivation by L-NAME, we analyzed the markers for fibroblast expansion and myofibroblast differentiation as well as the collagen density in mammary tissues after the drug treatment. To detect fibroblast expansion, we stained mammary tissues for the fibroblast marker, vimentin. As expected, L-NAME-treated mammary tissues showed a dramatic increase in periductal vimentin levels compared to control and L-arginine-treated mammary tissues (Figure 4A(a–f),B). To test for myofibroblast differentiation, we stained these tissues for α–SMA, the marker for contractile cells, namely, myofibroblasts, smooth muscle and myoepithelial cells of the mammary gland [37,38]. In control and L-arginine-treated mammary tissues, α–SMA expression was only seen at the periphery (i.e., basal layer) of the ducts, indicating that they were myoepithelial cells. Conversely, in L-NAME-treated mammary tissues, α–SMA expression was also elevated in periductal fibroblasts, aside from α–SMA-positive myoepithelial cells, indicating the emergence of myofibroblasts in the periductal region (Figure 4A(a–c),(g–i),B). The major function of myofibroblasts is to produce a large amount of collagen fibers, increasing ECM stiffness [38,39]. To test for an increase in stromal collagen by L-NAME treatment, we visualized collagen fibers using the second harmonic generation (SHG) technique. We indeed saw a dramatic increase in periductal collagen in L-NAME-treated tissues compared to control and L-arginine-treated tissues (Figure 4A(a–c),B) [23].

To test whether such fibrotic response of L-NAME treatment could be recapitulated in vitro, we applied the drug to 3D mono- and co-cultures and analyzed fibrogenic markers. As expected, neither 3D mono-cultured epithelia nor fibroblasts exhibited a fibrogenic response to L-NAME treatment. In 3D mono-cultured mammary epithelia, vimentin was not detectable for any of the treatments. Conversely, the majority of epithelial cells were α–SMA-positive, indicating myoepithelial cells, for all the treatments. Nevertheless, in L-NAME-treated colonies, which were composed of higher cell densities than control and L-arginine-treated colonies, the expression pattern of α–SMA had lost the basal polarity and pervaded the whole structures (Figure 4A(m–o),B). In 3D mono-cultured fibroblasts, on the other hand, vimentin levels were high, while α–SMA levels were low, for all the treatments (Figure 4A(s–u),B). Consistently, neither mono-cultured epithelia nor fibroblasts showed an increase in SHG signals by L-NAME treatment (Figure 4A(p–r),(v–x),B).

Unlike 3D mono-cultures, 3D co-cultures exhibited strong fibrogenic responses to L-NAME treatment. In co-cultured fibroblasts, both vimentin and α–SMA levels were highly elevated by L-NAME treatment, indicating their differentiation into myofibroblasts. In co-cultured epithelia, conversely, α–SMA expression became undetectable by L-NAME treatment, indicating the loss of myoepithelial cells—a frequent event in breast cancer progression (Figure 4A(c,f,a’,d’),B) [40]. Interestingly, the invasive phenotype of L-NAME-treated epithelia in co-cultures was even more pronounced than that of L-NAME-treated mammary glands that retained α–SMA expression at the periphery, indicating the persistent presence of myoepithelial cells (Figure 4A(c,i,a’). Consistently, SHG signals in 3D co-cultures were dramatically elevated by L-NAME treatment (Figure 4A(e’–g’),B). These results demonstrate that 3D co-cultures, but not 3D mono-cultures, exhibited in vivo-like fibrogenic responses to L-NAME treatment.

### 3.4. NO Deprivation Induces Epithelial-to-Mesenchymal Transition upon TGF-*β* Activation in 3D Co-Cultures of Mammary Epithelial Cells and Fibroblasts in a Manner that Resembles the In Vivo Response

As demonstrated above, L-NAME treatment elicited strong fibrogenic responses in mammary tissues, and this phenomenon was recapitulated in 3D co-cultures, but not in 3D mono-cultures (Figure 4). Interestingly, we observed that L-NAME-treated epithelia in both mammary tissues and 3D co-cultures showed a large increase in VIM levels (Figure 4A(c,f,a’,d’),B). We thus tested the possibility that these epithelia had undergone epithelial-to-mesenchymal transition (EMT)—a hallmark of invasive cancer progression. EMT is characterized by the increase in mesenchymal markers (e.g., VIM) and loss of epithelial markers (e.g., E-cadherin) [41]. In particular, Type 2 EMT is directly associated with tissue/organ fibrosis, where epithelial cells serve as precursors of fibroblasts/myofibroblasts [42]. We thus co-stained drug-treated mammary tissues for E-cadherin (E-CAD) vs. VIM to determine their relative abundance. As expected, control and L-arginine-treated mammary epithelia showed high levels of junctional E-CAD, but low levels of VIM. On the other hand, in L-NAME-treated epithelia, E-CAD expression was nearly abrogated, whereas the VIM level was highly elevated, indicating the occurrence of EMT (Figure 5A(a–i),B). As the mechanistic basis for the occurrence of EMT, in particular, Type 2 EMT, we tested for the involvement of TGF-β, a major inducer of this process [43,44]. In fact, L-NAME-treated mammary epithelia as well as periductal fibroblasts showed a dramatic increase in phospho-SMAD3, an indicator for the activation of TGF-β signaling (Figure 5A(j–I),B) [23,45].

To test whether the induction of Type 2 EMT by L-NAME treatment could also be observed in vitro, we applied the drug to 3D mono- and co-cultures and analyzed the same EMT markers. Neither 3D mono-cultured epithelia nor fibroblasts exhibited Type 2 EMT by L-NAME treatment. In 3D mono-cultured mammary epithelia, the levels of junctional E-CAD remained high, whereas VIM expression remained undetectable, for all the treatments (Figure 5A(m–r),B). Although phospho-SMAD3 was highly elevated in L-NAME-treated mono-cultured epithelia (Figure 5A(s–u),B), this was not sufficient to induce EMT. In 3D mono-cultured fibroblasts, conversely, E-CAD was virtually absent, while VIM expression remained high, for all the treatments. As expected, phospho-SMAD3 expression was absent in mono-cultured fibroblasts for all the treatments (Figure 5A(v–a’),B).

On the other hand, 3D co-cultures showed a clear occurrence of the in vivo-like EMT by L-NAME treatment. In co-cultured epithelia, L-NAME treatment abrogated E-CAD expression, but highly elevated VIM levels (Figure 5A(b’–j’),B). Consistently, L-NAME treatment largely elevated phospho-SMAD3 levels not only in epithelia (LAMA3-positive), but also in fibroblasts (LAMA3-negative) of co-cultures (Figure 5A(k’–m’),B). These results demonstrate that the induction of Type 2 EMT requires the crosstalk of TGF-β signaling between epithelia and fibroblasts, making 3D co-cultures, but not 3D mono-cultures, the best suited method to reproduce EMT in vitro.

### 3.5. NO Deprivation Impairs the Apico–Basal Polarity and Induces Invasive Phenotype of Mammary Epithelia in 3D Co-Cultures of Mammary Epithelial Cells and Fibroblasts

In addition to the expression of distinct molecular markers we analyzed above (Figure 5), EMT often manifests as the loss of apico–basal polarity, a major gatekeeper for tumorigenesis [42,46,47]. To test this possibility, we determined the expression patterns of commonly used polarity markers in mammary tissues and 3D cultures after drug treatment. The mammary gland is composed of a bi-layer of epithelia: the apical layer of luminal epithelial cells (positive for CK8/18 and GM130) and the basal layer of myoepithelial cells (positive for CK14 and α6 integrin) (Figure 1) [47,48,49,50]. As expected, control and L-arginine-treated mammary glands showed the well segregated bi-layers of CK8-positive luminal epithelia vs. CK14-positive myoepithelia at a 1:1 ratio [23,25,51]. In contrast, L-NAME-treated mammary glands, including intraductal papillomas (precancerous lesions), were predominantly composed of luminal epithelia (Figure 6A(a–c),B)[23]. This result demonstrates that L-NAME treatment obliterated the bi-layer system necessary for the apico–basal polarity of mammary glands.

We then tested whether L-NAME-mediated loss of polarity could also be observed in 3D mono- and co-cultures. Three-dimensional mono-cultures of mammary epithelia have been widely utilized to generate apico–basally polarized colonies, termed acinus-like structures, which closely resemble mammary glands [52]. Such 3D colonies, however, are not composed of a bilayer of luminal epithelia and myoepithelia as in mammary glands, but instead composed of an undefined mixture of both cell types. Thus, to determine the polarity, 3D colonies are often stained for apical (e.g., Golgi apparatus protein GM130) vs. basal polarity markers (integrin α 6: ITGA6), instead of apical vs. basal epithelial cells [33,53]. Consistent with previous reports, control and L-arginine-treated mono-cultured epithelia formed well-polarized acini that showed clear segregation of the apical (GM130) vs. basal polarity markers (ITGA6) [52]. Such segregation of polarity markers, however, was abolished by L-NAME treatment (Figure 6A(d–f),B) [23,34]. On the other hand, in 3D mono-cultured fibroblasts, the apical marker GM130 (Golgi apparatus protein) was detected in all the cells, whereas the basal marker ITGA6 was absent, demonstrating the lack of apico–basal polarity (Figure 6A(g–I),B).

In 3D co-cultures, control and L-arginine-treated mammary epithelia exhibited apico–basal polarity, whereas L-NAME-treated epithelia showed the loss of polarity, in a manner similar to mono-cultured epithelia (Figure 6A(j–o),B). Conversely, 3D co-cultured fibroblasts lacked apico–basal polarity for all the samples, also in a manner similar to mono-cultured fibroblasts (Figure 6A(g–I),B). These results demonstrate that our 3D co-cultures could serve as substitutes for 3D epithelial mono-cultures to recapitulate the in vivo-like apico–basal polarity of epithelia and its abrogation by L-NAME treatment.

The above observations altogether attested to the advantages of 3D co-cultures over 3D mono-cultures. Three-dimensional co-cultures exhibited the in vivo-like responses of epithelial morphology/polarity to L-NAME treatment in a manner similar to 3D mono-cultures (Figure 3 and Figure 6). However, 3D co-cultures were also able to reproduce the in vivo-like fibrogenesis and EMT under L-NAME treatment, whereas 3D mono-cultures failed to do so (Figure 4 and Figure 5). These findings strongly suggest that our 3D co-cultures would be better suited than 3D mono-cultures for use in the deep mechanistic studies of the causal relationship between fibrosis and cancer.

## 4. Discussion

Cumulative evidence suggests that excessive collagen accumulation and tissue fibrosis play pivotal roles in cellular transformation and tumor initiation [24,54,55,56]. Provenzano et al. demonstrated that crossing mammary tumor-prone mice (MMTV-PyMT) with mice overexpressing collagen I exacerbates tumor formation [22]. Furthermore, Kuperwasser et al. demonstrated that the co-transplantation of precancerous breast cells with genetically engineered CAFs (fibroblasts overexpressing HGF and/or TGF-β1) into the cleared mammary fat pads of immuno-deficient mice induced mammary tumors [20]. To understand the mechanistic bases of such phenomena, two separate studies by Pickup et al. and Mouw et al. demonstrated that ECM stiffness, as the result of the elevated collagen density, could upregulate the integrin-mediated focal adhesions. This, in turn, promotes oncogenic Ras, Myc and PI3K signaling and plasticity, leading to cellular transformation [37,57]. Furthermore, Elosegui-Artola et al. revealed that activated focal adhesions transmit forces to the nucleus through cytoskeletal connections, which could flatten the nucleus and stretch nuclear pores. This allows for the nuclear entry of YAP transcription factor, which activates the expression of genes involved in proliferation and plasticity, while inhibiting apoptotic genes [58,59].

Such experimental results are in line with epidemiological studies reporting the cancer-causing effects of fibrogenic agents such as asbestos and silica [60,61], as well as chronic conditions linked to fibrosis such as obesity and diabetes [62,63]. In addition, certain chemotherapeutic agents, such as bleomycin, gemcitabine and methotrexate, are found to induce organ fibrosis, which could worsen cancer patients’ outcomes [56]. Given that fibrogenic agents are now considered as major cancer-causative factors [64], the development of a robust method allowing for studies of the causal linkage between fibrosis and cancer would help augment the universal efforts to reduce cancer incidence.

Previously, we reported that the deprivation of NO is a fibrogenic and carcinogenic factor for the mammary tissue [23]. In the mammary tissue, physiological levels of NO play important roles in the development and functions of mammary glands [23,34,65,66,67]. During lactation, NO is even secreted into the breast milk as an essential component for neonatal immunity and growth [68]. On the other hand, NO production is downmodulated in many types of chronic conditions, such as aging [69], diabetes [70], obesity [71], cardiovascular disease [72,73] and cancer [23,74]. In fact, these chronic conditions frequently succumb to the formation of tissues/organs fibrosis, which is directly linked to increased cancer risk [75,76,77]. NO deficiency in disease conditions is largely attributed to the deficiency of the essential cofactor of NO synthase (NOS), tetrahydrobiopterin (BH_4_), which is extremely sensitive to oxidative degradation. The loss of BH_4_ causes the dissociation of the NOS dimer, disabling the canonical enzymatic function to produce NO [23,78,79,80,81].

In the present report, we tested whether such debilitating effects of NO deprivation could be recapitulated in our novel organotypic 3D co-cultures. This technique utilizes the discontinuous ECM system to culture parenchymal epithelia and stromal fibroblasts separately in the discrete, tissue-compliant ECM (epithelia in laminin-rich BM vs. fibroblasts in collagen-rich interstitial matrix), while allowing their paracrine interactions without direct physical interactions [24]. This new co-culture system is particularly useful for modeling glandular tissues, where the epithelium is enclosed by the laminin-rich BM to prevent its direct contact with the surrounding collagen I-rich interstitial membrane harboring fibroblasts (Figure 1A and Figure 2A). On the other hand, most, if not all, 3D co-culture systems previously developed utilize the continuous (a single type) matrix (collagen or collagen–Matrigel 1:1 mixture) (Figure 2A) [82,83,84,85] and are useful for recapitulating cancer cell invasion where the BM is lost.

Through the present study, we sought to test for the potential utility of our novel 3D co-culture system in the mechanistic studies of the linkage between fibrosis and cancer. We co-cultured mammary epithelia and mammary fibroblasts under the treatment of vehicle (control), L-arginine (NOS agonist) and L-NAME (NOS antagonist). The phenotypes of co-cultures were compared with those of drug-treated mouse mammary glands and 3D mono-cultured mammary epithelia or fibroblasts. L-NAME treatment of mammary glands induced TGF-β activation, triggering fibrogenic signals indicated by fibroblast expansion, myofibroblast differentiation (i.e., fibroblast activation) and increased collagen and vimentin levels. Such fibrogenic effects of L-NAME were well recapitulated in our 3D co-cultures, but not in 3D mono-cultures that lack epithelial-to-stromal interactions (Figure 4). Accordingly, the fibrogenic effects of L-NAME gave rise to Type 2 EMT (EMT associated with fibrogenesis) in mammary glands as well as 3D co-cultures, but not in 3D mono-cultures (Figure 5). On the other hand, morphogenetic effects of L-NAME that abolished the apico–basal polarity of mammary gland epithelia were reproduced in both 3D mono-cultures and co-cultures (Figure 6). Thus, the major advantage of our 3D co-culture over 3D mono-cultures is their ability to demonstrate the causal relationship between fibrogenesis and carcinogenesis as observed in vivo.

L-NAME is an unreactive L-arginine analog, serving as a competitive inhibitor of enzymes that utilize L-arginine as the substrate. In humans, NOS and arginase are the major enzymes targeted by L-NAME, where NOS has the higher affinity. At low concentrations (<5 mM), L-NAME only inhibits NOS activity, whereas at higher concentrations, it could also inhibit arginase [86,87,88]. Arginase is the enzyme that converts L-arginine to ornithine and urea. It is implicated in pro-tumor functions, and its inhibition is reported to suppress cancer cell proliferation [86]. Thus, the biological effects of L-NAME depend on the concentration, leading to somewhat conflicting observations [89,90,91,92,93,94,95]. For example, Pershing et al. applied 180 mg/kg of L-NAME to animals of a lung cancer model and observed tumor regression [89], suggesting that L-NAME at this concentration inhibited both NOS and arginase. On the other hand, Reis et al. applied 20 mg/kg of L-NAME to animals of a leukemia model and observed the increased tumor growth [95], suggesting that L-NAME here inhibited only NOS. Reis et al.’s report is in line with our finding that the treatment of wild-type animals with 20 mg/kg of L-NAME caused the formation of precancerous mammary lesions [23]. This was also consistent with our previous observation that 2.5 mM of L-NAME induces the loss of polarity and aberrant proliferation of normal mammary epithelia in 3D cultures [34].

While the present study specifically focuses on the utility of the 3D co-culture system for mammary epithelia and mammary fibroblasts, this system could potentially be extended to include additional stromal cells. Recently, various types of organotypic 3D co-culture systems have been developed. These include low fluid shear suspension culture [96], microvasculature model [97], 3D co-culture of epithelial cells, fibroblasts and leukocytes [98], 3D co-culture of adipocytes and macrophages [99] and 3D culture of adipose stromal-vascular fraction [100]. The discontinuous ECM co-culture system described in this report could be utilized in combination with these new techniques to more closely mimic the in vivo tissue conditions and physiological responses.

## 5. Conclusions

Overall, our results demonstrate that this new 3D co-culture system could better recapitulate the physiological responses of glandular tissues toward fibrogenic and carcinogenic agents than 3D mono-cultures. Such an advantage would justify the utility of this co-culture system as a robust in vitro model for the deep mechanistic studies of the causal relationship between fibrosis and cancer.

## Figures and Tables

**Figure 1 cancers-13-02815-f001:**
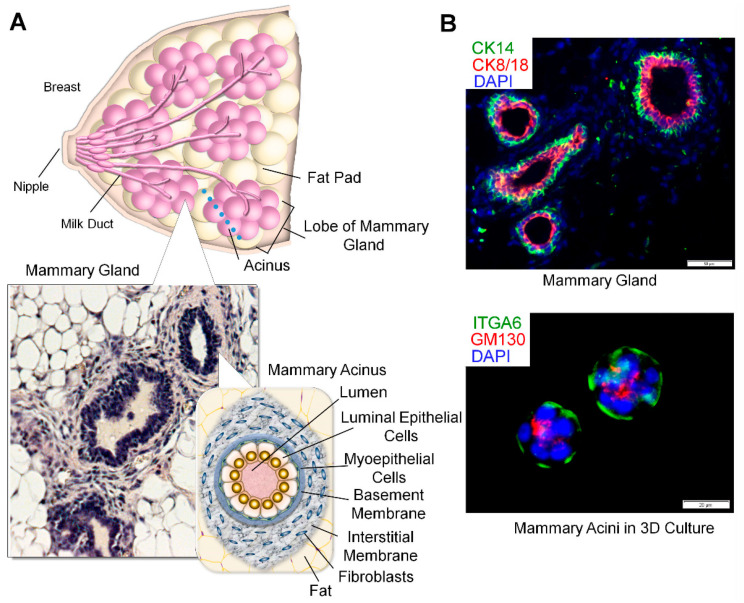
The architecture of mammary glands. (**A**) (Top) Exocrine glandular structure of the mammary gland embedded in the fat pad and connective tissue. (Bottom left) Cross-sectioned mammary gland stained with hematoxylin and eosin. (Bottom right) Structure of mammary acinus, milk-producing unit, composed of luminal epithelial cells and myoepithelial cells surrounded by the basement membrane (BM). Each acinus is embedded in the interstitial membrane (connective tissue) and fat tissue. (**B**) (Top) Cross-sectioned mammary gland stained with antibodies against cytokeratin 14 (green, basal cells) and cytokeratin 8/18 (red, luminal cells). (Bottom) Mammary acini formed by non-malignant human MECs, HMT-3522 S1, in 3D culture with Matrigel and stained with antibodies against α6 integrin (green, basal marker) and GM130 (red, apical marker). Nuclei were counterstained with DAPI (blue).

**Figure 2 cancers-13-02815-f002:**
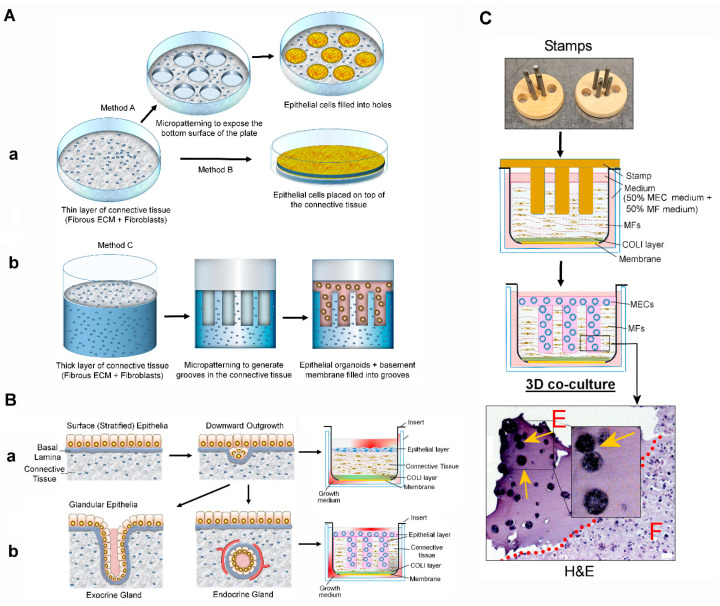
Schemes of organotypic 3D co-cultures for reconstructing surface epithelium vs. glandular epithelium. (**A**) Different organotypic 3D co-culture systems. (**a**) Fibroblasts are embedded in collagen matrix, and collagen matrix is micropatterned to expose the plastic substratum into which epithelial cells are seeded (Method A). Epithelial cells are grown on top of the fibroblast–collagen layer (Method B). (**b**) Collagen matrix is micropatterned to generate indentations into which epithelial organoids/basement membrane mixture is transferred (Method C). (**B**) (**a**) Structure of surface epithelium. (Left) The surface epithelia cover the underlying connective tissue. (Middle) Epithelial cells outgrow into the connective tissue. (Right) Organotypic 3D co-culture mimics the surface epithelia. (**b**) Structure of glandular epithelium. (Left) Outgrowing epithelia form invagination, which become duct and exocrine glands. (Middle) Once the outgrowing epithelia are separated from the surface epithelia, they form endocrine glands. (Right) Organotypic 3D co-culture mimics the glandular epithelia. (**C**) Scheme of organotypic 3D co-culture of primary mammary fibroblasts (MFs) and mammary epithelial cells (MECs). (Top panel) Image of custom-made stamps. (Second and third panels) Scheme of organotypic MF/MEC co-culture. (Bottom) Hematoxylin and eosin-stained paraffin-embedded section of co-culture. E: Epithelial cells; F: Fibroblasts. Scale bars: 50 μm.

**Figure 3 cancers-13-02815-f003:**
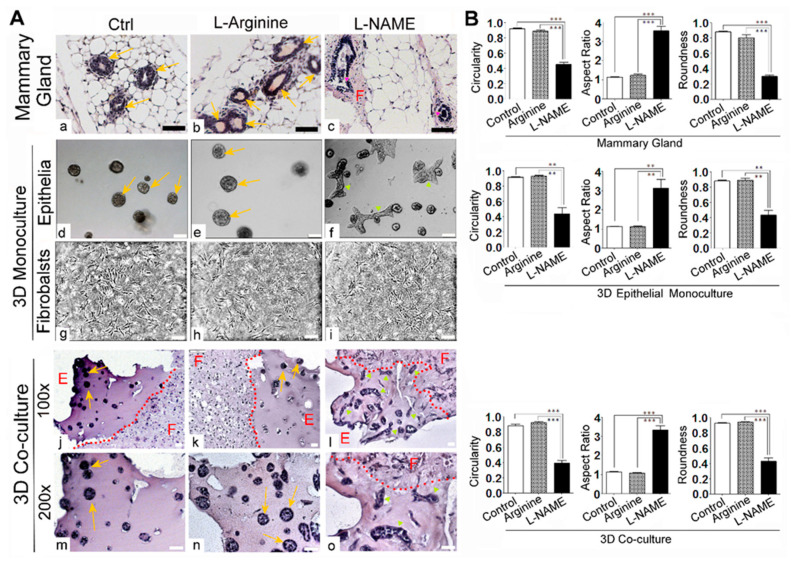
Treatment with nitric oxide synthase inhibitor L-NAME caused the aberrant morphologies of mammary gland epithelia and fibroblasts in vivo as well as in organotypic 3D co-cultures. (**A**) Representative micrographs of drug-treated samples. (**a**–**c**) Mouse mammary glands treated with control (vehicle), L-arginine (20 mg/kg, nitric oxide synthase agonist) or L-NAME (20 mg/kg, nitric oxide synthase inhibitor) for 6 weeks. Tissues were paraffin-embedded, sectioned and stained with hematoxylin and eosin. Note the formation of multiple intraductal papillomas as well as the expansion of periductal fibroblasts in L-NAME-treated glands. Three-dimensional mono-cultures of MECs (**d**–**f**) and MFs (**g**–**i**) treated with the same drugs (control: vehicle, L-arginine (2.5 mM) or L-NAME (2.5 mM)) for 2 weeks. Note the formation of elongated colonies in L-NAME-treated culture. (**j**–**l**) 100× magnification images of 3D organotypic co-cultures treated with the same drugs for 1 week (MECs and MFs were also pre-treated with the drugs for 1 week before being placed in co-cultures). Co-cultures were treated with control (vehicle), L-arginine (2.5 mM) or L-NAME (2.5 mM) throughout the whole culturing period. Co-cultures were harvested, paraffin-embedded, sectioned and stained with H&E. Red dotted lines separate MECs (E, glandular epithelial structures) in Matrigel vs. MFs (F, connective tissues) in collagen/Matrigel matrix. (**m**–**o**) 200× magnification images of co-cultures. Note the elongated structures of MEC colonies and the irregular shapes and orientation of MFs in L-NAME-treated sample. Arrows: mammary acini; pink arrowheads: intraductal papillomas of mammary glands; yellow arrowheads: elongated 3D colonies. Scale bars: 50 μm. (**B**) Quantification of shape descriptors (Circularity, Aspect Ratio and Roundness) of epithelia in micrographs of (**A**) Top row: mammary glands. Middle row: 3D mono-cultures. Bottom row: 3D co-cultures. Circularity = 4π*area/perimeter^2. Aspect Ratio = major_axis/minor_axis. Roundness = 4*area/(π*major_axis^2). Data are presented as the arithmetic mean ± SEM. ** *p* < 0.01; and *** *p* < 0.001.

**Figure 4 cancers-13-02815-f004:**
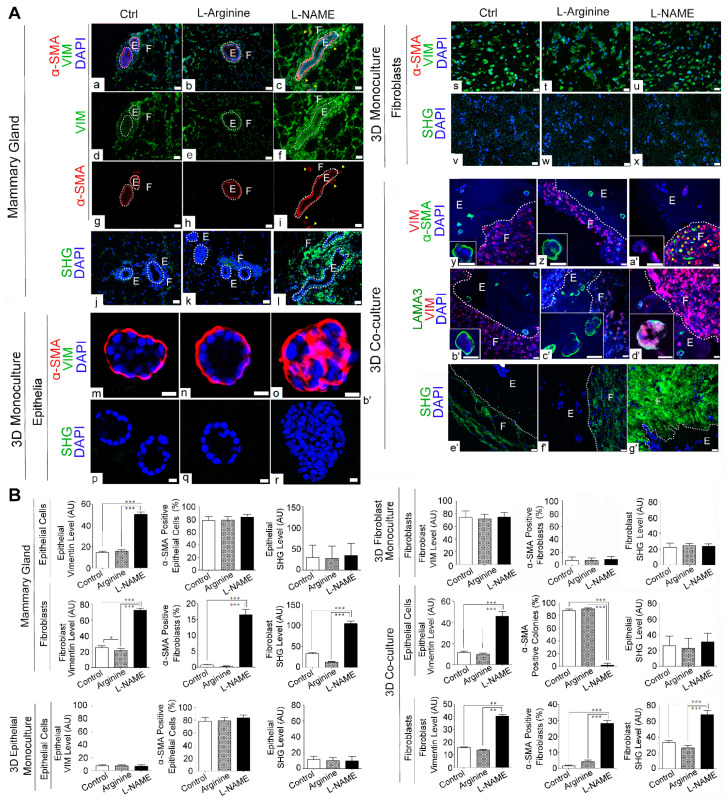
L-NAME treatment triggers fibrogenic signaling in mammary glands as well as in 3D co-cultures. (**A**) Representative micrographs of drug-treated mouse mammary glands (**a**–**l**); 3D mono-cultured epithelia (**m**–**r**); fibroblasts (**s**–**x**); and 3D co-cultures (**y**–**g’**) as in Figure 3 (Control, L-arginine-treated and L-NAME-treated). Co-stained for vimentin (VIM, marker for fibroblasts) and α-smooth muscle actin (α-SMA, marker for myoepithelia and myofibroblasts) ((**a**–**i**), (**m**–**o**), (**s**–**u**), (**b’**–**d’**)). Yellow arrowheads: α-SMA-positive periductal fibroblasts (myofibroblasts). Co-stained for VIM (fibroblasts) and laminin α3 (LAMA3, basement membrane/hemidesmosome marker) (**b’**–**d’**). Second harmonics generation imaging (SHG) to visualize collagen fibers (**j**–**l**), (**p**–**r**), (**v**–**x**), (**e’**–**g’**). Insets show representative epithelial colonies. White dotted lines separate MECs (E, glandular epithelial structures) vs. MFs (F, connective tissues). Note the dramatic increase in α-SMA-positive myofibroblasts and VIM-positive epithelia as well as SHG signals, by L-NAME treatment in mammary glands and 3D co-cultures, but not in 3D mono-cultured epithelia and fibroblasts. Scale bars: 20 μm. (**B**) Quantification of signal intensity of different markers shown in micrographs of A in epithelia vs. fibroblasts. Data are presented as the arithmetic mean ± SEM. ** *p* < 0.01; and *** *p* < 0.001.

**Figure 5 cancers-13-02815-f005:**
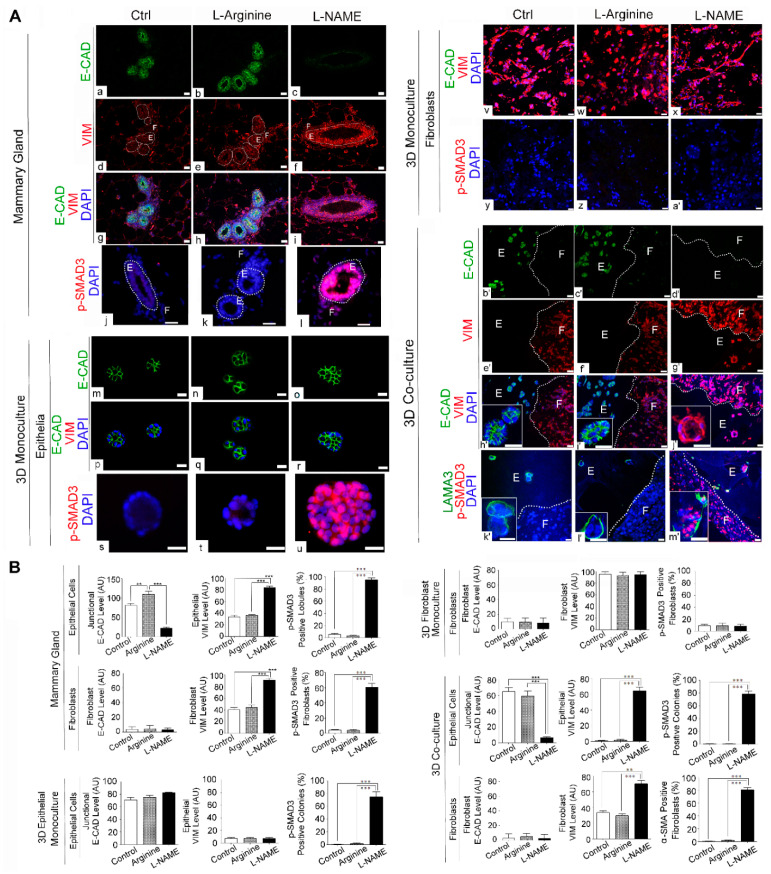
L-NAME treatment elevates TGF-β signals and induces epithelial-to-mesenchymal transition in mammary glands as well as in 3D co-cultures. (**A**) Representative micrographs of drug-treated mouse mammary glands (**a**–**l**); 3D mono-cultured epithelia (**m**–**u**); fibroblasts (**v**–**a’**); and 3D co-cultures (**b’**–**m’**) as in Figure 3 (Control, L-arginine-treated and L-NAME-treated). Co-stained for E-cadherin (E-CAD, epithelial marker) and vimentin (VIM, fibroblast marker) ((**a**–**i**), (**m**–**r**), (**v**–**x**), (**b’**–**j’**)). Stained for p-SMAD3, an indicator of TGF-β signaling ((**j**–**l**), (**s**–**u**), (**y**–**a’**), (**v’**–**m’**)). (Panels j–l, s, t were adapted from Ren et al. [23] with permission of Springer Nature.) Insets show representative epithelial colonies. White dotted lines separate MECs (E, glandular epithelial structures) vs. MFs (F, connective tissues). Note that L-NAME treatment downmodulated E-CAD expression, but upregulated VIM expression in the epithelia, indicating epithelial-to-mesenchymal transition (EMT). L-NAME-induced EMT coincided with elevated TGF-β signaling (increase in p-SMAD3), the major activator of EMT as well as fibrogenesis. (**B**) Quantification of signal intensity of different markers shown in micrographs of A in epithelia vs. fibroblasts. Data are presented as the arithmetic mean ± SEM. ** *p* < 0.01; and *** *p* < 0.001.

**Figure 6 cancers-13-02815-f006:**
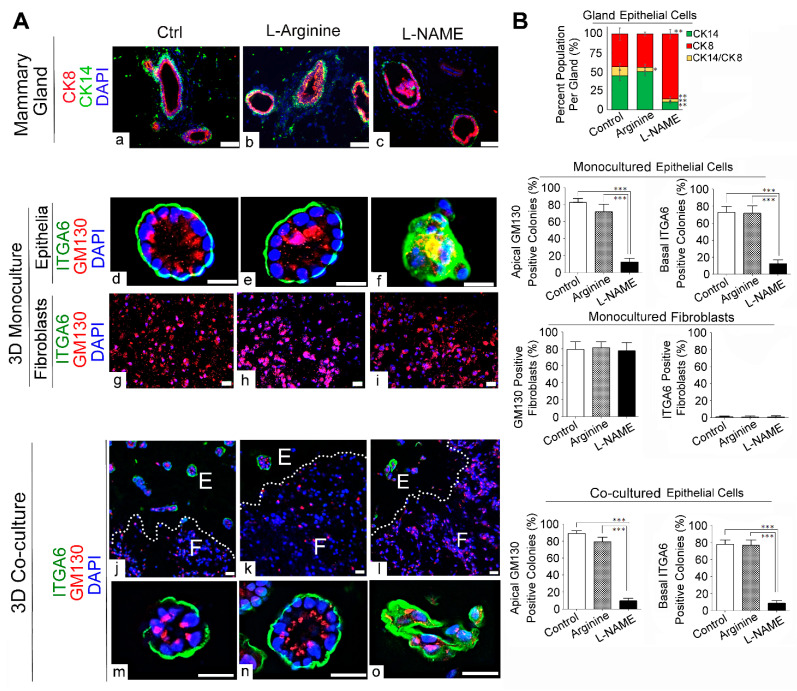
L-NAME treatment induced the outgrowth of luminal epithelial cells and loss of apico–basal polarity in mammary acini in vivo and in 3D mono- and co-cultures. (**A**) Representative micrographs of drug-treated samples. (**a**–**c**) Drug-treated mouse mammary glands as in Figure 3 (Control, L-arginine- and L-NAME-treated) were stained with antibodies against CK8 (luminal epithelial cell marker) and CK14 (myoepithelial marker). Note the dramatic increase in CK8 level and decrease in CK14 level in L-NAME-treated glands. (**d**–**i**) Drug-treated 3D mono-cultures of MECs (**d**–**f**) and fibroblasts (**g**–**i**) as in Figure 3 were stained with GM130 (apical marker) and integrin α6 (ITG6, basal marker). Note the aberrant apical polarity in L-NAME-treated MEC colonies (**f**). (**j**–**o**) Drug-treated organotypic 3D co-cultures of MECs (E, glandular epithelial structures) and MFs (F, connective tissues) as in Figure 3 were stained with GM130 (lumen marker) and ITG6 (basal marker). Note the loss of apical marker and disorganized basal marker in MECs in L-NAME-treated co-cultures ((**l**,**o**). Scale bars: 50 μm. (**B**) Quantification of signal intensity of different markers analyzed in micrographs of (**A**) (Top) Percentage of CK14 basal, CK8 (luminal) and CK14/CK8-positive cells in each gland. Data are presented as the arithmetic mean ± SEM. * *p* < 0.05; ** *p* < 0.01; and *** *p* < 0.001.

## Data Availability

The data presented in this study are available on request from the corresponding author.

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
