# Peer review of "Loss of Nitric Oxide Induces Fibrogenic Response in Organotypic 3D Co-Culture of Mammary Epithelia and Fibroblasts—An Indicator for Breast Carcinogenesis"

_cancers, 2021, doi:10.3390/cancers13112815_

Round 1
Reviewer 1 Report
This present study, submitted by Ren at al, evaluates the utility of a novel 3D co-culture model to assess NO depletion on mimicking pre-cancerous mammary transformation. Their results reveal their model performs better than mono-cultured fibroblast/epithelia in recapitulating the phenomenon. Their results serve as a proof of concept for a model of in vitro culture that can be thus used to screen other (non-NO) fibrotic agents for their effects on the 3D cocultures.
Overall, this is a very strong paper. While their findings may have limited utility when generalized to other fields, they are still relevant to the current development of 3D in vitro models in breast cancer research, and with some work, may be generalized to other models as well. Unfortunately, there is no screening of compounds that occurs in this manuscript, which would greatly strengthen the work; however, even without this, the manuscript is quite strong.
I identified just a few minor things:
1. Simple summary needs work on grammar throughout, some awkward phrasing.
2. Line 321 - “a frequent event in breast cancer progression” requires citation. 3. Line 470, bold on citations to be removed (typo) 4. Line 512 - “Pershing et al” requires a space (typo)Author Response
Reviewer 1
Comments and Suggestions for Authors
Overall, this is a very strong paper. While their findings may have limited utility when generalized to other fields, they are still relevant to the current development of 3D in vitro models in breast cancer research, and with some work, may be generalized to other models as well. Unfortunately, there is no screening of compounds that occurs in this manuscript, which would greatly strengthen the work; however, even without this, the manuscript is quite strong.
Response: We truly appreciate the reviewer’s supportive comments. Please note that all the corrections made according to the reviewer’s comments are highlighted in yellow in the revised manuscript.
Specific Comments:
I identified just a few minor things:
- Simple summary needs work on grammar throughout, some awkward phrasing.
Response: We appreciate the reviewer’s comment. We have corrected the grammatical errors of Simple Summary section
- Line 321 - “a frequent event in breast cancer progression” requires citation.
- Line 470, bold on citations to be removed (typo) (now line 499)
- Line 512 - “Pershing et al” requires a space (typo) (now line 540)
Response: We appreciate the reviewer’s comment. We have addressed all the points raised by the reviewer.

Reviewer 2 Report
Summary
The manuscript titled ‘Loss of Nitric Oxide Induces Fibrogenic Response in Organotypic 3D Co-culture of Mammary Epithelia and Fibroblasts—an Indicator for Breast Carcinogenesis’ investigates the ability of using a 3D co-culture model for screening potential fibrogenic and carcinogenic agents in the context of breast tissues. They tested their model by treating it with L-NAME, an agent that is known to deplete nitric oxide by depleting NO synthase that in turn results in fibrosis and tumorigenesis in mammary glands. They were able to show that their 3D co-culture model was able to show different hallmarks of fibrosis and tumorigenesis that includes increased collagen production, epithelial-mesenchymal transition, and loss of apical basal polarity.
Strengths
The study is well designed and well written. The greatest strength of the article is the that the authors compared their results from the 3D co-culture model with 3D monoculture models as well as in vivo mouse models. Additionally, the authors have designed experiments to address different aspects of fibrogenesis as well as tumorigenesis in their 3D model. There are clear explanations on how different criteria were evaluated like circularity, etc.
Weakness
Specific recommendations are provided below.
- Their 3D model shows many similarities with the in vivo models but there are differences as well. The authors should mention about this and elaborate on it the way they have for the 3D monoculture model.
- In figure 4B, the % of a-SMA positive epithelial cells in the 3D monoculture shows no difference among the treatments. However, in figure 4A, the same condition appears to show more cells and the area of expression is more. The authors do note in their description that ‘Conversely, α–SMA expression, indicating myoepithelial cells, was high for all the treatments, although in L-NAME-treated colonies the expression pattern had lost the basal polarity and pervaded the whole structures’. The authors should find a way to demonstrate this in their graphs also in order to avoid any confusion to the reader.
- In figure 6, there appears to be a mismatch between some of the images (A) and the graphs (B).
- The graph for the monocultured epithelial cells shows significantly reduced basal ITGA6 expression in the L-NAME condition, but the corresponding image shows good expression of ITGA6 (green color). There is a change in the localization pattern that they have addressed, but it is not clear why there is a reduced expression shown in the bar graph.
- The graphs for the Co-cultured epithelial cells shows significantly reduced apical GM130 expression in the L-NAME condition, but the corresponding figure does show expression of GM130 (red color). There is a change in the localization pattern that they have addressed, but it is not clear why there is a reduced expression shown in the bar graph.
- Minor revision: The margins from the Materials and Methods onwards is different than the Introduction. Please correct the formatting.
These revisions will strengthen the manuscript and make it easier for the readers to appreciate the importance of the findings.
Author Response
Reviewer 2
Comments and Suggestions for Authors
Strengths
The study is well designed and well written. The greatest strength of the article is the that the authors compared their results from the 3D co-culture model with 3D monoculture models as well as in vivo mouse models. Additionally, the authors have designed experiments to address different aspects of fibrogenesis as well as tumorigenesis in their 3D model. There are clear explanations on how different criteria were evaluated like circularity, etc.
Response: We truly appreciate the reviewer’s supportive comments. Please note that all the corrections made according to the reviewer’s comments are highlighted in yellow in the revised manuscript.
Weakness
Specific recommendations are provided below.
- Their 3D model shows many similarities with the in vivo models but there are differences as well. The authors should mention about this and elaborate on it the way they have for the 3D monoculture model.
Response: We appreciate and agree with the reviewer’s comment. We did found significant difference in the expression of α–SMA after L-NAME treatment between the in vivo mammary glands and 3D co-cultures. In 3D co-cultures, L-NAME-treated epithelia completely lost α–SMA expression, indicating the loss of myoepithelial cells. In contrast, L-NAME-treated mammary glands retained the peripheral expression of α–SMA. Such difference indicates that L-NAME-treated 3D co-cultures demonstrated more invasive phenotype of epithelia than the corresponding in vivo condition.
To emphasize this difference, we added a sentence, “Interestingly, the invasive phenotype of L-NAME-treated epithelia in co-cultures was even more pronounced than that of L-NAME-treated mammary glands that retained α–SMA expression at the periphery, indicating the persistent presence of myoepithelial cells (Figures 4A, c, i, a’)”.
- In figure 4B, the % of α-SMA positive epithelial cells in the 3D monoculture shows no difference among the treatments. However, in figure 4A, the same condition appears to show more cells and the area of expression is more. The authors do note in their description that ‘Conversely, α–SMA expression, indicating myoepithelial cells, was high for all the treatments, although in L-NAME-treated colonies the expression pattern had lost the basal polarity and pervaded the whole structures’. The authors should find a way to demonstrate this in their graphs also in order to avoid any confusion to the reader.
Response: We apologize for the confusion. We believe that such confusion comes from the fact that the numbers of epithelial cells per colony in mono-cultures are different between control/arginine-treated samples (Figure 4A, m, n) vs. L-NAME-treated sample (Figure 4A, o). Despite that, the percentages (~80%) of α-SMA-positive cells per colony, as shown in the corresponding graph in B, are the same for all the treatments.
To resolve such a confusion, we’ve revised the sentences describing this phenomonon to “Conversely, the majority of epithelial cells were α–SMA-positive, indicating myoepithelial cells, for all the treatments. Nevertheless, in L-NAME-treated colonies, which were composed of higher cell densities than control and L-arginine-treated colonies, the expression pattern of α–SMA had lost the basal polarity and pervaded the whole structures (Figures 4A, m-o, 4B)”.
- In figure 6, there appears to be a mismatch between some of the images (A) and the graphs (B).
Response: We apologize for the confusion. We rearranged the Figure 6 to make it easier for the reader to associate the figure panels in A with the corresponding graphs in B.
- The graph for the monocultured epithelial cells shows significantly reduced basal ITGA6 expression in the L-NAME condition, but the corresponding image shows good expression of ITGA6 (green color). There is a change in the localization pattern that they have addressed, but it is not clear why there is a reduced expression shown in the bar graph.
Response: For this analysis, we focused on the “basal localization” of ITGA6 (basal marker), but not “the total expression level”. (ITGA6 is a cell surface receptor that binds the ECM protein laminin, and the total expression level won’t significantly change by different treatments.) We determined the percentage of colonies that had “basal” ITGA6 expression (i.e., basally polarized colonies). Since the majority of L-NAME-treated colonies had lost their polarity, the number of colonies with basal ITGA6 expression was reduced.
- The graphs for the Co-cultured epithelial cells shows significantly reduced apical GM130 expression in the L-NAME condition, but the corresponding figure does show expression of GM130 (red color). There is a change in the localization pattern that they have addressed, but it is not clear why there is a reduced expression shown in the bar graph.
Response: This question is analogous to the above question. For this analysis, we focused on the “apical localization” of GM130 (apical marker), but not “the total expression level”. (GM130 is a structural protein of the Golgi apparatus, and the total expression level won’t change by different treatments.) We determined the percentage of colonies that had “apical” GM130 expression (i.e., apically polarized colonies). Since the majority of L-NAME-treated colonies had lost their polarity, the number of colonies with apical GM130 expression was reduced.
- Minor revision: The margins from the Materials and Methods onwards is different than the Introduction. Please correct the formatting.
Response: We appreciate the reviewer’s comment. We’ve corrected the formatting of the margin.
These revisions will strengthen the manuscript and make it easier for the readers to appreciate the importance of the findings.
Response: We truly appreciate the reviewer’s comments!

Reviewer 3 Report
The manuscript describes the results of the experimental study of the experimental treatment of native and in vitro reconstructed mammary glands by L-NAME, the blocker of nitric oxide synthase. The manuscript is interesting as it provides new data that can contribute to the better understanding of the early stages of breast cancer (and, possibly, some other cancers) development. Importantly, the study validates the original 3D co-culture model of mammary glands vs the matching animal model, both in intact state and after the treatment. Using the pharmacological NO-deprivation approach, the authors provide convincing evidence in favour of the induced pro-fibrotic effect in malignant progression depicted by morphological changes of the mammary glands, EMT and polarity loss. The results of the study are well-presented and discussed, and, I believe, worth publishing.
From the scientific side, I have only a few minor suggestions:
- I recommend to edit the Simple Summary and present it in plain language. The role of this section of the manuscript is to deliver the message of the study to the general public. Please, try to simplify the text. Maybe, a kind of explanation of the title of the paper, could work for that. Please try to explain why it is important to study cancer and fibrosis in 3D co-cultures (co-existence of epithelial cancer cells with the organ-specific stroma? The role of stroma in cancer treatment resistance? Etc.), why nitric oxide is important, what did the study revealed? In the current text, the listed agents (asbestos, silica and bleomycin) look like a random choice, their causative role in cancer is non-specific, and there is no much cohesiveness with the further manuscript-told story. The logical links between them and nitric oxide are also unclear from this simple presentation, but maybe quite important.
- For the whole study, I think that it’s a little bit of overclaiming that the presented methodology is ready and suitable for screening of various agents in 3D co-culture systems. The main issue that concerns me is the term “screening” which implies fast, reproducible, high-throughput assays. The current study proves that the presented co-culture system is very well adapted for the deep studies of cancer and fibrosis mechanisms, and, possibly, to explore the mechanistic aspects of the experimental treatments/interventions. However, for the screening purposes, this 3D co-culture system may be too complex, and not reproducible enough.
- Other minor corrections & comments:
- Row 42, “Dense collagenous stroma, namely, desmoplasia..” should be “Formation of the dense collagenous stroma…” - because “desmoplasia” is a process.
- Figure 1, b (bottom) – please, add the explanation of the blue staining (DAPI?).
- Figure 3 and related text: possibly, some clarification is needed to explain why the observation/in vitro culture periods were different/were chosen like indicated in different groups. Also, do you think that the same dosage of L-NAME in 3D mono- and co-cultures acts in the same way?
I congratulate the authors with a very interesting and high-quality work and recommend this paper to be published after minor corrections.
Author Response
Reviewer 3
Comments and Suggestions for Authors
The manuscript describes the results of the experimental study of the experimental treatment of native and in vitro reconstructed mammary glands by L-NAME, the blocker of nitric oxide synthase. The manuscript is interesting as it provides new data that can contribute to the better understanding of the early stages of breast cancer (and, possibly, some other cancers) development. Importantly, the study validates the original 3D co-culture model of mammary glands vs the matching animal model, both in intact state and after the treatment. Using the pharmacological NO-deprivation approach, the authors provide convincing evidence in favour of the induced pro-fibrotic effect in malignant progression depicted by morphological changes of the mammary glands, EMT and polarity loss. The results of the study are well-presented and discussed, and, I believe, worth publishing.
Response: We truly appreciate the reviewer’s supportive comments. Please note that all the corrections made according to the reviewer’s comments are highlighted in yellow in the revised manuscript.
Specific Comments:
From the scientific side, I have only a few minor suggestions:
- I recommend to edit the Simple Summary and present it in plain language. The role of this section of the manuscript is to deliver the message of the study to the general public. Please, try to simplify the text. Maybe, a kind of explanation of the title of the paper, could work for that. Please try to explain why it is important to study cancer and fibrosis in 3D co-cultures (co-existence of epithelial cancer cells with the organ-specific stroma? The role of stroma in cancer treatment resistance? Etc.), why nitric oxide is important, what did the study revealed? In the current text, the listed agents (asbestos, silica and bleomycin) look like a random choice, their causative role in cancer is non-specific, and there is no much cohesiveness with the further manuscript-told story. The logical links between them and nitric oxide are also unclear from this simple presentation, but maybe quite important.
Response: We appreciate the reviewer’s comment. We have revised the Simple Summary section in an attempt to incorporate all the reviewer’s suggestions. Because of the space restriction (150 words), we succinctly introduced all these points raised by the reviewer.
- For the whole study, I think that it’s a little bit of overclaiming that the presented methodology is ready and suitable for screening of various agents in 3D co-culture systems. The main issue that concerns me is the term “screening” which implies fast, reproducible, high-throughput assays. The current study proves that the presented co-culture system is very well adapted for the deep studies of cancer and fibrosis mechanisms, and, possibly, to explore the mechanistic aspects of the experimental treatments/interventions. However, for the screening purposes, this 3D co-culture system may be too complex, and not reproducible enough.
Response: We appreciate and agree with the reviewer’s comment. As suggested by the reviewer, we have changed the phrase “utilized for screening for fibrogenic and carcinogenic agents” to “utilized for the deep studies of the mechanistic link between fibrosis and cancer” throughout the manuscript.
- Other minor corrections & comments:
- Row 42, “Dense collagenous stroma, namely, desmoplasia..” should be “Formation of the dense collagenous stroma…” - because “desmoplasia” is a process.
Response: We appreciate the reviewer’s comment. We have corrected this section to “Formation of the dense collagenous stroma” according to the reviewer’s suggestion.
- Figure 1, b (bottom) – please, add the explanation of the blue staining (DAPI?).
Response: We appreciate the reviewer’s comment. We have included the sentence “Nuclei were counterstained with DAPI (blue).” At the bottom of the revised Figure 1 legend.
- Figure 3 and related text: possibly, some clarification is needed to explain why the observation/in vitro culture periods were different/were chosen like indicated in different groups. Also, do you think that the same dosage of L-NAME in 3D mono- and co-cultures acts in the same way?
Response: We appreciate the reviewer’s comment. 3D mono-cultured mammary epithelia and fibroblasts were treated with drugs for 2 weeks. In case of 3D co-cultures, conversely, mammary epithelia and fibroblasts were pre-treated with the drugs for 1 week, and after being placed in co-cultures, they were further treated with the same drugs for 1 week. Thus, the total duration of drug treatment was the same between mono-cultures and co-cultures. To clarify this point, we have included the sentence “MECs and MFs were also pre-treated with the drugs for 1 week before being placed in co-cultures” in the revised Figure 3 legend.
Regarding the question of whether the same dosage of L-NAME in 3D mono- and co-cultures acts in the same way, we would not have a definite answer. However, we would assume that the effective concentrations of L-NAME in both culture systems would be similar based on the following two factors:
- L-NAME (N(G)-Nitro-L-arginine methyl ester) is an analog of the amino acid L-arginine. It should be able to diffuse through the ECM (both laminin-rich and collagen I-rich matrices) [1], and the concentration should reach equilibrium in both mono- and co-cultures systems within hours.
- The total duration of L-NAME treatment was the same (2 weeks) for both mono- and co-cultures, where the media with the drug were replaced every other day.
I congratulate the authors with a very interesting and high-quality work and recommend this paper to be published after minor corrections.
Response: We truly appreciate the reviewer’s supportive comments!
References:
- Fan, D.; Creemers, E.E.; Kassiri, Z. Matrix as an interstitial transport system. Circulation research 2014, 114, 889-902, doi:10.1161/circresaha.114.302335.
